# Differential methylation between ethnic sub-groups reflects the effect of genetic ancestry and environmental exposures

Joshua M Galanter[1,2,3]*‡, Christopher R Gignoux[4], Sam S Oh[1,2,3], Dara Torgerson[2], Maria Pino-Yanes[5,6], Neeta Thakur[1], Celeste Eng[1], Donglei Hu[1], Scott Huntsman[1], Harold J Farber[7], Pedro C Avila[8], Emerita Brigino-Buenaventura[9], Michael A LeNoir[10], Kelly Meade[11], Denise Serebrisky[12], William Rodríguez-Cintrón[13], Rajesh Kumar[14], Jose R Rodríguez-Santana[15], Max A Seibold[16], Luisa N Borrell[17], Esteban G Burchard[1,2]*†, Noah Zaitlen[1]*†

[1]Department of Medicine, University of California, San Francisco, United States; [2]Department of Bioengineering and Therapeutic Sciences, University of California, San Francisco, United States; [3]Department of Epidemiology and Biostatistics, University of California, San Francisco, San Francisco, United States; [4]Department of Genetics, Stanford University, Stanford, United States; [5]Hospital Universitario Nuestra Señora de Candelaria, Tenerife, Spain; [6]CIBER de Enfermedades Respiratorias, Instituto de Salud Carlos III, Madrid, Spain; [7]Department of Pediatrics, Baylor College of Medicine and Texas Children's Hospital, Houston, Texas; [8]Division of Allergy and Immunology, Feinberg School of Medicine, Northwestern University, Chicago, Illinois; [9]Kaiser Permanente-Vallejo Medical Center, Vallejo, United States; [10]Bay Area Pediatrics, Oakland, United States; [11]Department of Pediatrics, Children's Hospital and Research Center, Oakland, United States; [12]Jacobi Medical Center, Bronx, United States; [13]Veterans Caribbean Health System, San Juan, United States; [14]Division of Allergy and Immunology, The Ann and Robert H Lurie Children's Hospital of Chicago, Chicago, United States; [15]Centro de Neumología Pediátrica, San Juan, United States; [16]Center for Genes, Environment, and Health, Department of Pediatrics, National Jewish Health, Denver, United States; [17]Graduate School of Public Health and Health Policy, City University of New York, New York, United States

*For correspondence: galanter@protonmail.com (JMG); esteban.burchard@ucsf.edu (EGB); noah.zaitlen@ucsf.edu (NZ)

†These authors contributed equally to this work

Present address: ‡Genentech, South San Francisco, United States

Competing interests: The authors declare that no competing interests exist.

**Abstract** Populations are often divided categorically into distinct racial/ethnic groups based on social rather than biological constructs. Genetic ancestry has been suggested as an alternative to this categorization. Herein, we typed over 450,000 CpG sites in whole blood of 573 individuals of diverse Hispanic origin who also had high-density genotype data. We found that both self-identified ethnicity and genetically determined ancestry were each significantly associated with methylation levels at 916 and 194 CpGs, respectively, and that shared genomic ancestry accounted for a median of 75.7% (IQR 45.8% to 92%) of the variance in methylation associated with ethnicity. There was a significant enrichment ($p = 4.2 \times 10^{-64}$) of ethnicity-associated sites amongst loci previously associated environmental exposures, particularly maternal smoking during pregnancy. We conclude that differential methylation between ethnic groups is partially explained by the shared genetic ancestry but that environmental factors not captured by ancestry significantly contribute to variation in methylation.

**eLife digest** Whether a person develops a particular disease can depend on both genetic and environmental factors. Many studies have found that people of different races and ethnicities have different likelihoods of acquiring certain diseases. Race and ethnicity are social constructs; that is, they are not necessarily defined biologically. However, shared ancestry will produce genetic links between members of a group. In addition, members of an ethnic group often share a culture or environment that may influence their risk of disease. For example, the 'Mediterranean diet' inspired by the dietary habits of Southern Italians has been shown to reduce the risk of heart disease, diabetes and cancer.

The addition of chemical groups – such as methyl groups – to DNA strands can affect the activity of nearby genes. Methylation is controlled by both genetic and environmental factors, and altered patterns of DNA methylation are seen in some diseases. It is therefore an ideal biological process to study to determine how race/ethnicity and ancestry contribute to a person's susceptibility to disease.

Galanter et al. have now studied the patterns of methylation found in the blood of 573 people from diverse Latino ethnic sub-groups. The different groups displayed significantly different patterns of methylation at hundreds of locations across the genome. Genetic ancestry explained approximately 75% of the variation in methylation between the sub-groups. In addition, the methylation patterns at DNA locations known to be affected by environmental exposures – for example, by exposure to tobacco while in the womb – were disproportionately likely to be methylated differently in different sub-groups.

Now that more is known about the relative effects of race/ethnicity and genetic ancestry on methylation, the next step is to apply this knowledge to disease processes. This will help us to better understand the source of health disparities across different groups of people.

## Introduction

Race, ethnicity, and genetic ancestry have had a complex and often controversial history within biomedical research and clinical practice (*Risch et al., 2002*; *Cooper et al., 2003*; *Yudell et al., 2016*; *Burchard et al., 2003*; *Phimister, 2003*). For example, race- and ethnicity-specific clinical reference standards are based on population-based sampling on a given physical trait such as pulmonary function (*Hankinson et al., 1999*; *Quanjer et al., 2012*). However, because race and ethnicity are social constructs and poor markers for genetic diversity, they fail to capture the heterogeneity present within racial/ethnic groups and in admixed populations (*Borrell, 2005*). To account for these heterogeneities and to avoid social and political controversies, the genetics community has grouped individuals by genetic ancestry instead of race and ethnicity (*Yudell et al., 2016*). Indeed, recent work from our group and others have demonstrated that genetic ancestry improves diagnostic precision compared to racial/ethnic categorizations for specific medical conditions and clinical decisions (*Kumar et al., 2010*; *Udler et al., 2015*; *Nalls et al., 2008*).

However, racial and ethnic categories also reflect the shared experiences and exposures to known risk factors for disease, such as air pollution and tobacco smoke, poverty, and inadequate access to medical services, which have all contributed to worse disease outcomes in certain populations (*Nguyen et al., 2014*; *Evans and Kantrowitz, 2002*). Thus, it is unclear whether defining groups through genetic ancestry can capture these shared exposures. In this work we seek to explore the contributions of genetically defined ancestry and social, cultural and environmental factors to understanding differential methylation between ethnic groups.

Epigenetic modification of the genome through methylation plays a key role in the regulation of diverse cellular processes (*Smith and Meissner, 2013*). Changes in DNA methylation patterns have been associated with complex diseases, including various cancers (*Kulis and Esteller, 2010*), cardiovascular disease (*Udali et al., 2013*; *Kato et al., 2015*), obesity (*Bell et al., 2010*), diabetes (*Chambers et al., 2015*), autoimmune and inflammatory diseases (*Liu et al., 2013*), and

neurodegenerative diseases (*Lardenoije et al., 2015*). Epigenetic changes are thought to reflect influences of both genetic (*Bell et al., 2011*) and environmental factors (*Feil and Fraga, 2011*), and have been shown to vary between racial groups (*Barfield et al., 2014*). The discovery of methylation quantitative trait loci (meQTL's) across populations by Bell et al. established the influence of genetic factors on methylation levels in a variety of tissue types (*Bell et al., 2011*), with meQTL's explaining between 22% and 63% of the variance in methylation levels. Multiple environmental factors have also been shown to affect methylation levels, including endocrine disruptors, tobacco smoke (*Joubert et al., 2012, 2016*), polycyclic aromatic hydrocarbons, infectious pathogens, particulate matter, diesel exhaust particles (*Jiang et al., 2014*), allergens, heavy metals, and other indoor and outdoor pollutants (*Ho et al., 2012*). Psychosocial factors, including measures of traumatic experiences (*Chen et al., 2013*; *Ressler et al., 2011*; *van der Knaap et al., 2014*), socioeconomic status (*Lam et al., 2012*; *Borghol et al., 2012*), and general perceived stress (*Vidal et al., 2014*), also affect methylation levels. Since both genetic and environmental exposures affect methylation, this represents an ideal phenotype to explore the contributions of these two factors on differential methylation between ethnic groups.

In this work, we leveraged genome-wide methylation data in 573 Latino children of diverse Latino sub-ethnicities enrolled in the Genes-Environment and Admixture in Latino Americans (GALA II) study (*Oh et al., 2012*) whose genetic ancestry had been determined from dense genotyping arrays. This allowed us to explore the extent to which the differences in methylation between Latino sub-groups could be explained by their shared genetic ancestry. We found that many of the methylation differences associated with ethnicity could be explained by shared genetic ancestry. However, even after adjusting for ancestry, significant differences in methylation remained between the groups at multiple loci, reflecting social and environmental influences upon methylation.

Our findings have important implications for both the use of ancestry to capture biological changes and of race/ethnicity to account for social and environmental exposures. Epigenome-wide association studies in diverse populations may be susceptible to confounding due to environmental exposures in addition to confounding due to population stratification (*Michels et al., 2013*). The findings also have implications for the common practice of considering individuals of Latino descent, regardless of origin as a single ethnic group.

## Results

The study included 573 participants, the majority of whom self-identified as being either of Puerto Rican (n = 220) or Mexican origin (n = 276). *Table 1* displays baseline characteristics of the GALA II study participants with methylation data included in this study, stratified by ethnic subgroups (Puerto Rican, Mexican, Other Latino, and Mixed Latinos who had grandparents of more than one national origin). *Figure 1* shows the distribution of African, European, and Native American ancestry among the 524 participants with genomic ancestry estimates.

Methylation data used in this study has been previously made publicly available at the Gene Expression Omnibus at https://www.ncbi.nlm.nih.gov/geo/query/acc.cgi?acc= GSE77716 (*Rahmani et al., 2016*). Genotyping data has been deposited in dbGaP; link will be activated when the data becomes publicly available (Burchard, http://www.ncbi.nlm.nih.gov/gap/? term=phs001180).

### Global patterns of methylation

Differences in ethnicity and ancestry resulted in discernible patterns in the global methylation profile as demonstrated in a multidimensional scaling analysis (*Figure 2A*). As expected (*Houseman et al., 2012*; *Lam etal., 2012*), the first few principal coordinates are strongly correlated to imputed cell composition (*Figure 2B–C*). There are also significant associations of self-identified sub-ethnicity with PC2 (p-ANOVA = 0.003), PC3 (p-ANOVA = 0.004), PC6 (p-ANOVA = 0.0001), PC7 (p-ANOVA = 0.0003) (*Figure 3A*), and PC8 (p-ANOVA = 0.0003), after adjusting for age, sex, disease status, cell components, and technical laboratory factors (plate and position). Genetic ancestry was associated with PC3 (p=0.002), PC7 (p=0.0004) (*Figure 3B*) and PC8 (p=0.001) in a two degree of freedom ANOVA test, adjusting for age, sex, disease status, cell components, technical factors, and ethnicity. *Supplementary file 1A* summarizes the results of the simple correlation analysis of

**Table 1.** Baseline characteristics of GALA II participants with methylation data, stratified by ethnicity. Continuous variables are reported with inter-quartile range in brackets.

| | Mexican | Puerto rican | Mixed latino | Other latino |
|---|---|---|---|---|
| n | 276 | 220 | 16 | 61 |
| Males (%) | 125 (45.3%) | 127 (57.7%) | 6 (37.5%) | 28 (45.9%) |
| Age | 11.4 [9.3: 14.7] | 12.3 [10.4: 14.2] | 11.8 [10.7: 14.9] | 11.8 [10: 15.7] |
| Asthma cases (%) | 124 (44.9%) | 147 (66.8%) | 9 (56.3%) | 31 (50.8%) |
| | Ancestry (n = 524) | | | |
| African | 4.3% [2.9%: 6.0%) | 22.8% [16.6%: 29.4%) | 8.5% [5.6%: 19.2%) | 12.3% [6.3%: 25.8%) |
| Native **American** | 55.4% [44.5%: 65.7%) | 11.2% [9.8%: 13%) | 31.5% [20.9%: 45.6%) | 32.8% [10.4%: 49.3%) |
| European | 40.5% [29.9%: 50.2%) | 65.7% [59.2%: 71%) | 50.5% [44.6%: 57.6%) | 48.9% [40%: 58.5%) |
| | Recruitment Site | | | |
| Chicago | 140 (50.7%) | 15 (6.8%) | 11 (68.9%) | 15 (24.6%) |
| New York | 18 (6.5%) | 10 (4.5%) | 1 (6.3%) | 23 (37.7%) |
| Puerto Rico | 0 | 193 (87.7%) | 0 | 0 |
| San Francisco | 78 (28.3%) | 0 | 2 (12.5%) | 23 (37.7%) |
| Houston | 40 (14.5%) | 2 (0.9%) | 2 (12.5%) | 5 (8.2%) |
| | Cell Counts (estimated) | | | |
| Granulo cytes | 51.2% [46.0%: 55.7%) | 51.6% [46.8%: 57%) | 51% [43.6%: 57.2%) | 49.1% [43.8%: 55.8%) |
| Lympho cytes | 41.9% [36.9%: 46.6%) | 41.8% [36.9%: 46.5%) | 41.9% [36.1%: 51.6%) | 43.9% [36.8%: 49.6%) |
| Mono cytes | 7.1% [5.8%: 8.3%) | 6.74% [5.74%: 8.24%) | 6.6% [5.7%: 7.6%) | 7.4% [6.2%: 8.6%) |

methylation with ethnicity and ancestry, as well as the adjusted nested ANOVA models described above and the mediation results described below.

A mediation analysis (*Tingley et al., 2014*) revealed that the associations between ethnicity and PCs 3, 7, and eight were significantly mediated by Native American ancestry, which explained ~100% (95% CI: 37–100%, p=0.01) of PC3, 83% (95% CI 37–100%, p<0.001) of PC7 and 66% (95% CI: 25% to 100%, p<0.001) of PC8. Inclusion of Native American ancestry in the regression model of PCs 3, 7, and eight caused the ethnicity associations to be non-significant. However, the associations of ethnicity with PCs 2 and 6 were not explained by Native American, African or European ancestry (mediation p>0.05), suggesting that the ethnic differences in these principal components are associated with global methylation patterns not captured by the shared genetic ancestry of each ethnic group. When genetic ancestry was regressed on the methylation data with the principal coordinates recalculated using the residuals of the regression between methylation and ancestry, there was an association between ethnicity and PC6 (p-ANOVA = 0.003). However, there was no association with any of the other principal coordinates. These observations suggest that while shared genetic ancestry can explain over 50% of the association between ethnicity and global methylation patterns in three PC's, other non-genetic factors, such as environmental and social exposure differences associated with ethnicity influence methylation and are not captured by measures of genetic ancestry in two others.

## Epigenome-wide association of self-identified ethnicity

An epigenome-wide association study of self-identified ethnicity (see Materials and methods for details of ascertainment of ethnicity) and methylation identified a significant difference in methylation M-values between ethnic groups at 916 CpG sites at a Bonferroni-corrected significance level of

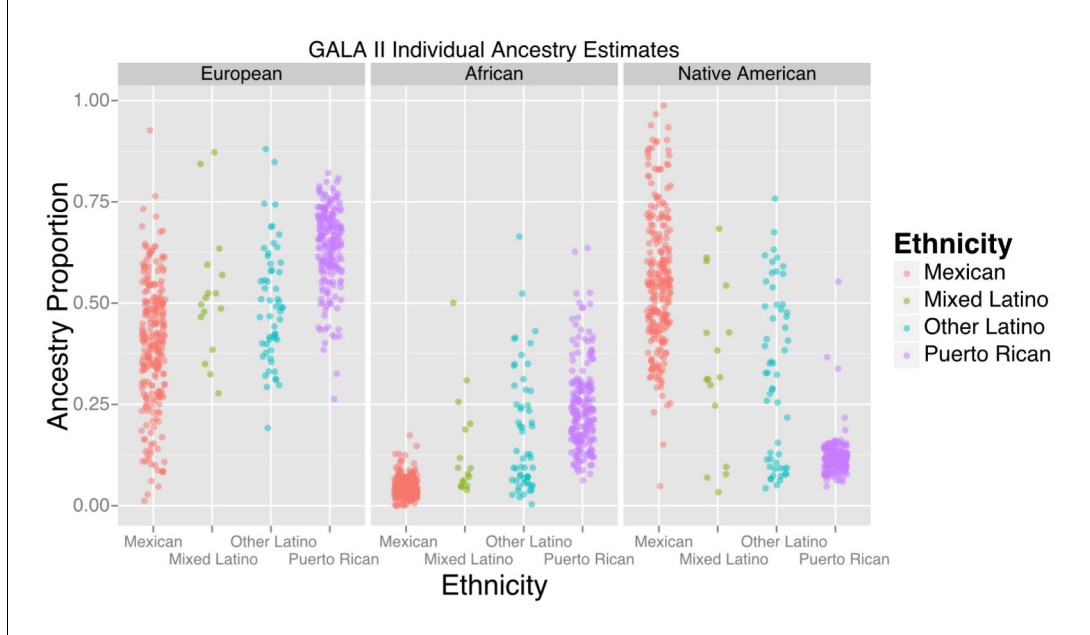

**Figure 1.** Ancestry estimates for GALA II participants, by ethnic group. Mexicans, on average, had a greater proportion of Native American ancestry than Puerto Ricans; Puerto Ricans had a greater proportion of European and African ancestry. Mixed and other Latinos were intermediate.

less than $1.6 \times 10^{-7}$ (*Figure 4A* and *Supplementary file 1B*). The most significant association with ethnicity occurred at cg12321355 in the ABO blood group gene (*ABO*) on chromosome 3 (p-ANOVA $6.7 \times 10^{-22}$) (*Figure 4B*). A two degree of freedom ANOVA test for genomic ancestry was also significantly associated with methylation level at this site (p=$2.3\times10^{-5}$) (*Figure 4C*), and when the analysis was stratified by ethnic sub-group, showed an association in both Puerto Ricans and Mexicans (p=0.001 for Puerto Ricans, p=0.003 for Mexicans). Although adjusting for genomic ancestry attenuated the effect of ethnicity, a significant association between ethnicity and methylation remained (p=0.04). Recruitment site, an environmental exposure proxy, was not significantly associated with methylation at this locus (p=0.5), suggesting that environmental differences associated with ethnicity beyond geography and ancestry are driving the association.

To determine the contribution of shared genetic ancestry and other factors associated with ethnicity, we repeated the analysis adjusting for ancestry. A significant association remained in 314 of the 834 (37.8%, p=$1.7\times10^{-183}$ for enrichment) CpG sites associated with ethnicity (*Figure 5A* and *Supplementary file 1B*) (82 sites were excluded because they demonstrated unstable coefficient estimates and inflated standard errors due to strong correlations between ethnicity and ancestry, especially Native American ancestry [see *Figure 1*]).

*Table 2* and *Figure 5b* show the proportion of variance explained by ethnicity, genomic ancestry, and their joint effect in the 916 CpG's associated with ethnicity, as well as the 314 CpG's that remained associated with ethnicity after adjustment for ancestry and the 520 CpG's whose association with ethnicity was no longer significant when ancestry terms were introduced into the model. Even after adjusting for genomic ancestry, ethnicity explained 1.7% (IQR 0.785% to 3.0%) but as much as 13.4% of the variance in methylation across these loci. Genomic ancestry explained a median of 4.2% (IQR 1.8% to 8.3%) of the variance in methylation at all loci associated with ethnicity and accounts for a median of 75.7% (IQR 45.8% to 92%) of the total variance in methylation explained jointly by ethnicity and ancestry (median of 6.8%, IQR 4.5% to 10.0%) (*Figure 5B*).

Ethnicity and ancestry jointly explained as much as 38.5% of the variance in methylation in one CpG (cg0966827) and there were 17 CpG's where ethnicity and ancestry jointly explain more 25% of the variance. Among the 314 CpG's that remained associated with ethnicity after adjustment for ancestry, ethnicity accounted for a larger share of the joint variance than genomic ancestry (3.5%, IQR 2.2% to 5.1% versus 1.8%, IQR 0.8% to 4.0%). We saw a moderate amount of correlation

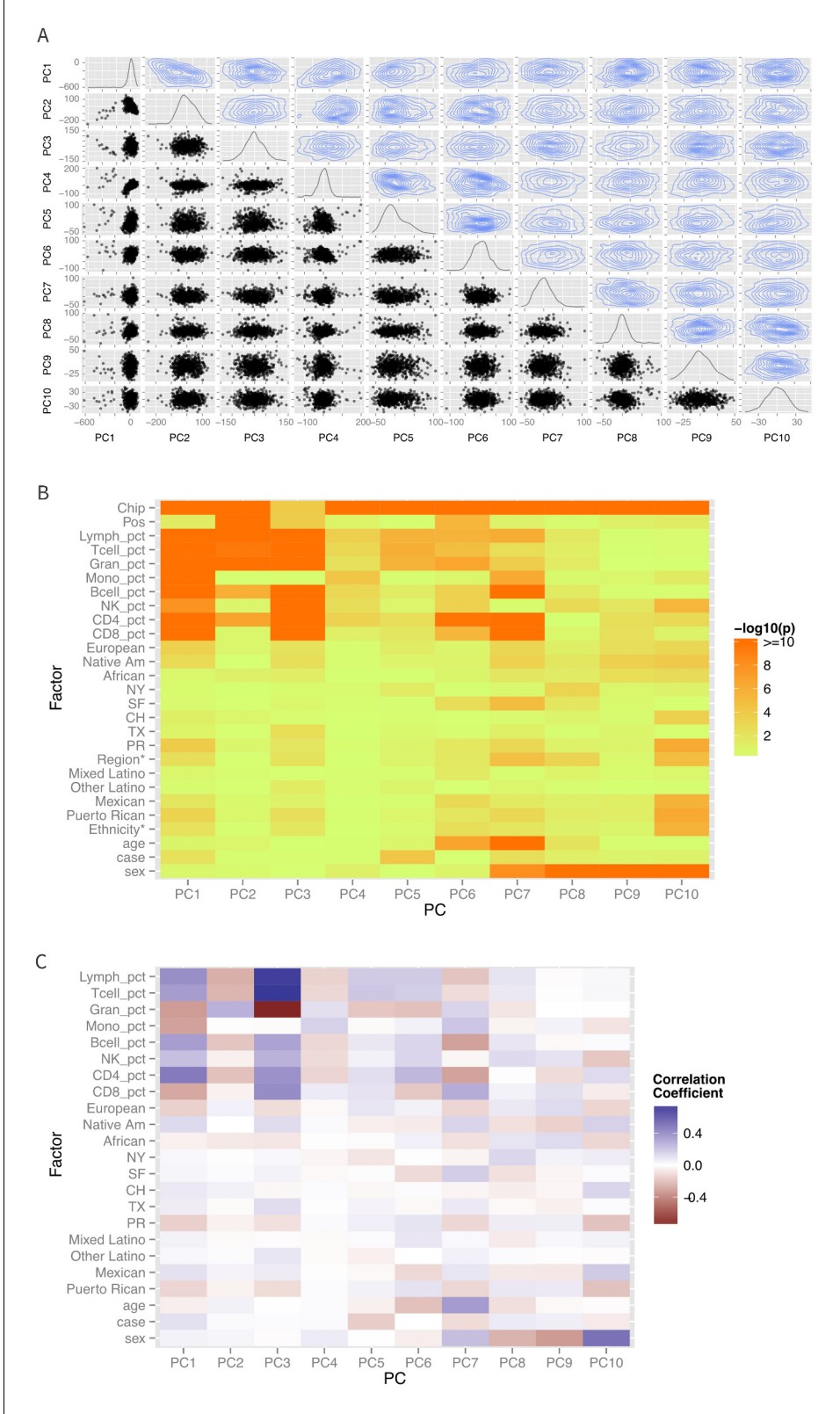

**Figure 2.** Patterns of global methylation. (**A**) Distribution of the first 10 principal coordinates of the methylation data. Plots in the diagonal show the univariate distribution; those in the lower left triangle show bivariate relationship between each pair of PCs, while those in the upper right show the bivariate density. (**B**) Bivariate or ANOVA associations between principal coordinates and technical factors (chip, position), cell counts, genetic ancestry

*Figure 2 continued on next page*

*Figure 2 continued*

(European, Native American, African), recruitment site (New York, NY, San Francisco, CA, Chicago, IL, Houston, TX, and Puerto Rico), demographic factors (ethnicity, age, sex), and case status. (C) Correlation coefficients between the various factors and principal coordinates.

between the 314 methylation sites associated with ethnicity after adjusting for ancestry (median $R^2$ of 0.044, IQR 0.01 to 0.13).

Sensitivity tests for departures from linearity, fine scale population substructure and the exclusion of the 16 participants who self-identified as 'Mixed Latino' sub-ethnicity, did not meaningfully affect

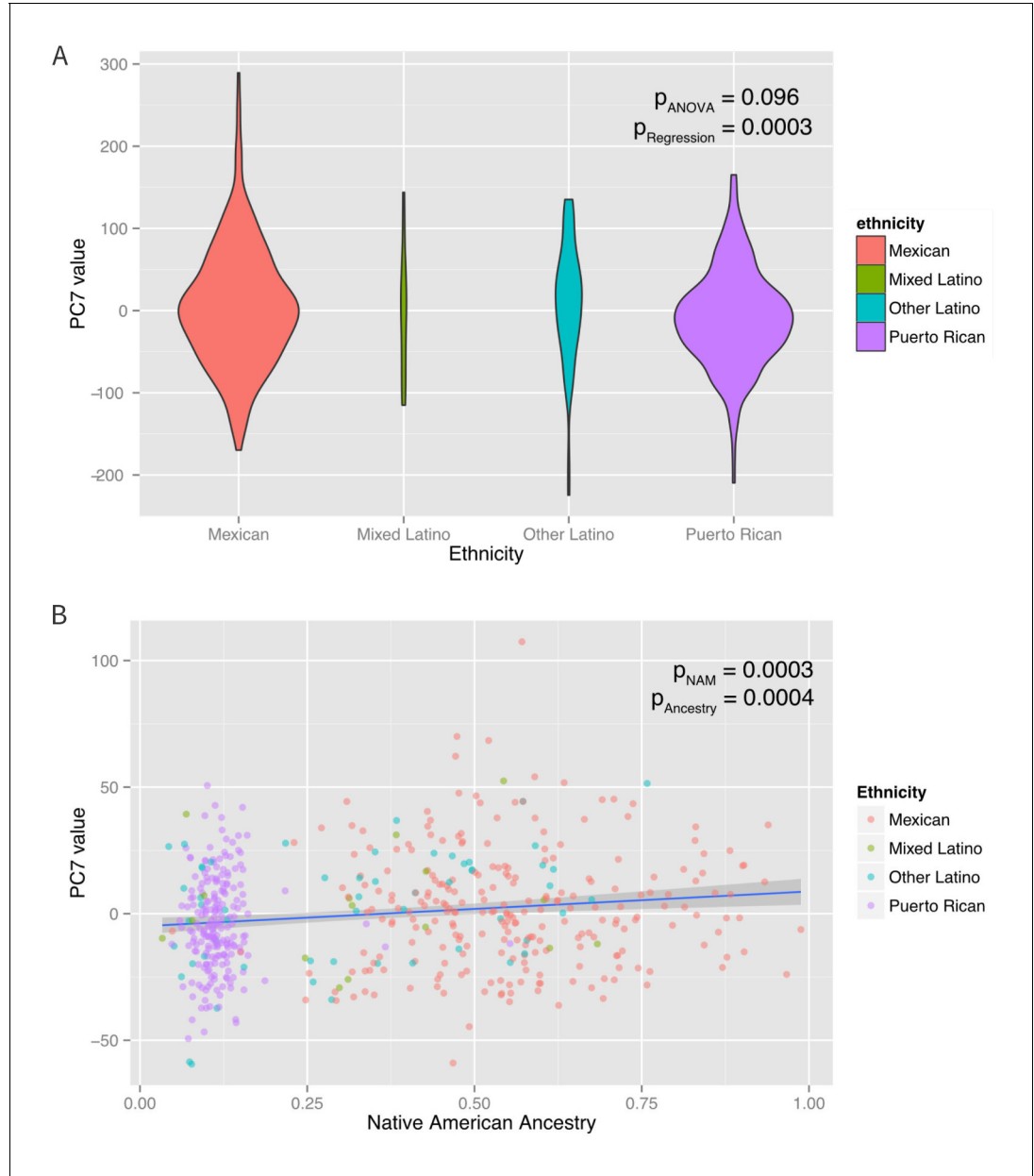

**Figure 3.** Associations between ethnicity, ancestry and global methylation. (**A**) Association between ethnicity and principal coordinate 7. (**B**) Association between Native American ancestry proportion and PC7, colored by ethnicity. Native American ancestry explains approximately 81% of the association between PC7 and ethnicity.

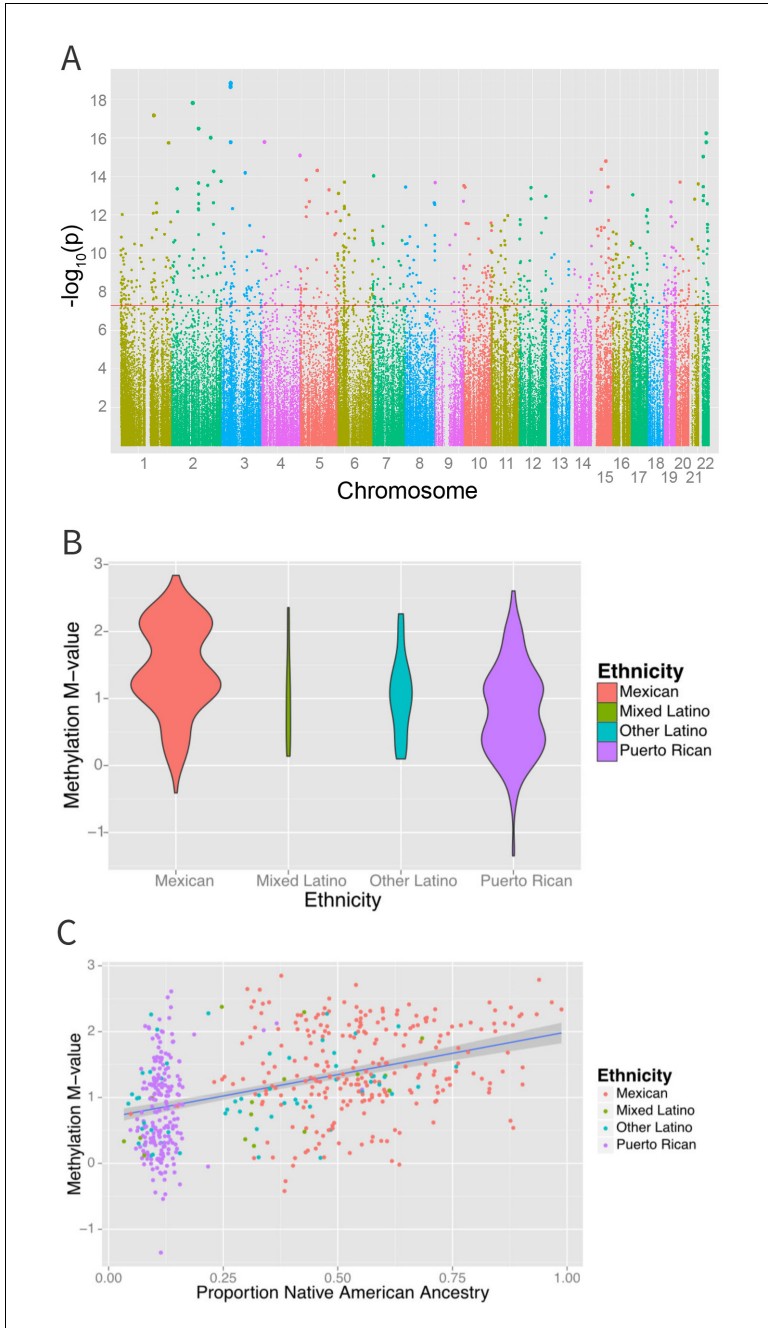

**Figure 4.** Associations between ethnicity and methylation (**A**) Manhattan plot showing the associations between ethnicity and methylation at individual CpG loci. (**B**) Violin plot showing one such locus, cg19145607. Mexicans are relatively hypermethylated compared to Puerto Ricans (p=1.4×10–19). (**C**) Plot showing the association between Native American ancestry at the locus and methylation levels at the locus colored by ethnicity; Native American ancestry accounts for 58% of the association between ethnicity and methylation at the locus.

our results (See *Supplementary file 1B–F*). To rule out any residual confounding due to recruitment sites, we conducted an additional analysis on the effect of recruitment site on methylation both for the overall study and for the Mexican participants (the largest study population in this analysis). We observed no significant independent effect of recruitment site suggesting that confounding due to recruitment region was limited, at least within the United States.

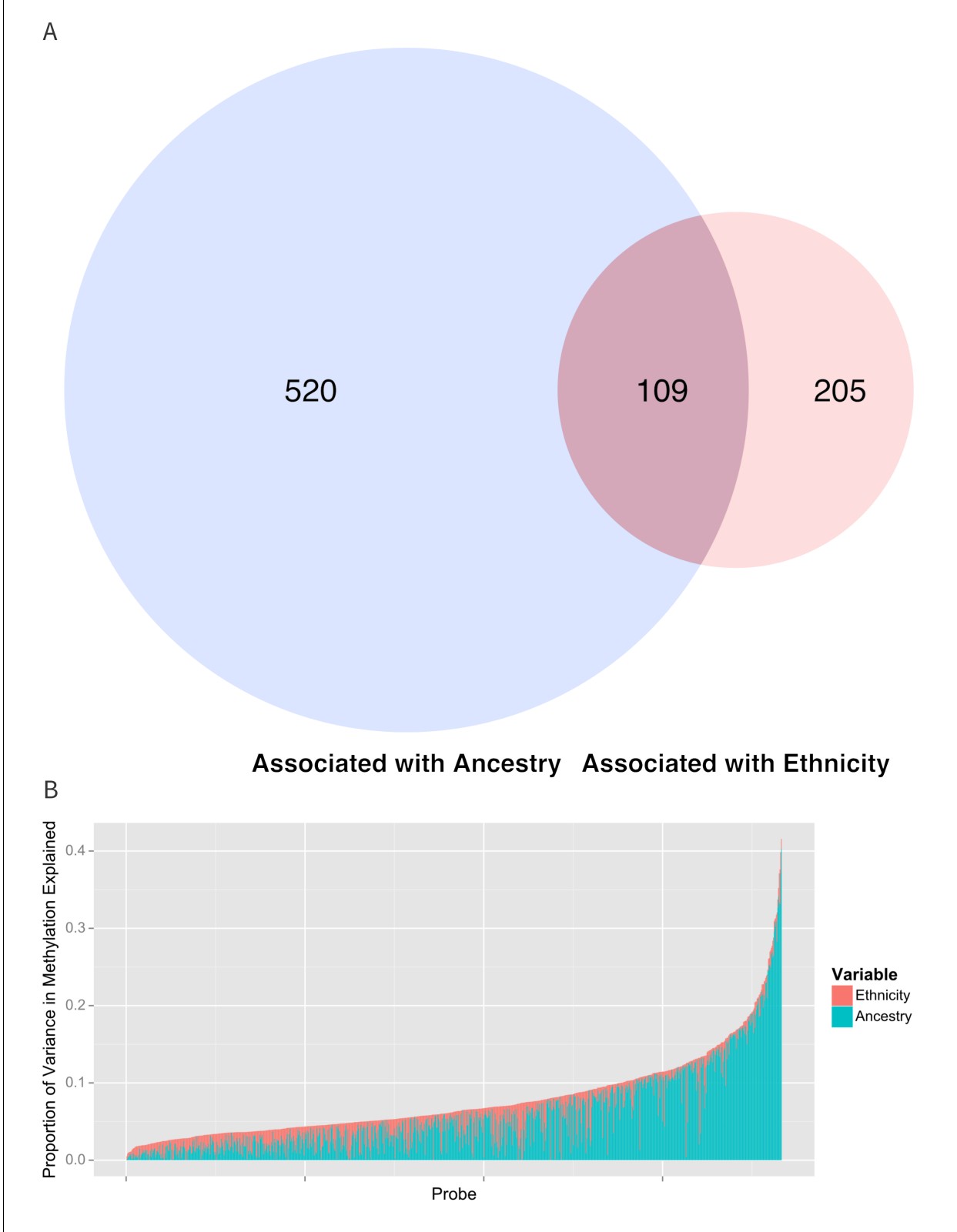

**Figure 5.** Relationship between genomic ancestry and the association between ethnicity and methylation. (**A**) Venn diagram showing the effect of adjustment for ancestry on the association between ethnicity and methylation. The components of the diagram represent the number of CpG's that remained associated with ethnicity after adjustment for ancestry and the number of CpG's that were associated with ancestry. (**B**) Relative proportion of variance in methylation explained by ethnicity and genomic ancestry across loci significantly associated with ethnicity. Mediation analysis of associations

*Figure 5 continued on next page*

*Figure 5 continued*

between ethnicity and methylation M-values for (**C**) Native American ancestry and (**D**) African ancestry. For simplicity, only significant mediation effects are shown.

To explore the effect of departures from a linear association between ancestry and methylation, we incorporated both higher order polynomials and cubic splines of ancestry into our models. We observed a significant departure from linearity ($p<0.05$) in only 26 (for splines) and 25 (for polynomials) of the 314 CpG's where an association between ethnicity and methylation remained after adjusting for ancestry; however, the association between ethnicity and methylation remained even after adjusting for non-linearity at all sites (*Supplementary file 1C,D*).

Environmental differences between geographic locations or recruitment sites are a potential non-genetic explanation for ethnic differences in methylation. We investigated the independent effect of recruitment site on methylation by analyzing the associations between recruitment site and individual methylation loci after adjusting for ethnicity. We did not find any loci significantly associated with recruitment site at a significance threshold of $1.6\times10^{-7}$. We then performed an analysis to assess the effect of recruitment sites on methylation stratified by ethnicity. We did not find any loci significantly associated with recruitment site and methylation among Mexican participants. We were underpowered to perform a similar analysis for Puerto Ricans because there were only 27 Puerto Rican participants recruited outside of Puerto Rico. To ensure that the absence of association in Mexicans was not due to the loss of power from the smaller sample size, we repeated our analysis of the association between ethnicity and ancestry randomly down-sampling to 276 participants to match the sample size in the analysis of geography in Mexicans. While down-sampling the study to this degree resulted in a loss of power, 128 methylation sites were still associated with ancestry. We conclude that recruitment site was unlikely to be a significant confounder of our associations between ethnicity and methylation and was not a significant independent predictor of methylation.

While most population substructure in Latinos would be expected to arise from differences in continental ancestry (*Galanter, 2012*; *Bryc et al., 2010*), there is evidence of finer scale (sub-continental) ancestry in Latino populations (*Moreno-Estrada et al., 2014*). We tested for the effect of fine scale substructure by calculating principal components for all participants with genotyping data using Eigensoft (*Patterson et al., 2006*). We found significant associations between principal components 3–10 (PC's 1 and 2 were almost perfectly collinear with ancestry, with an adjusted $R^2 > 0.998$ for all three ancestry proportions, and were therefore excluded) and ethnicity. We therefore added these 8 PC's to models of ethnicity and methylation, and found an association between these genetic PC's and methylation in 63/314 CpG's that had remained associated with ethnicity after adjusting for ancestry. Adjusting for higher order substructure in these CpG's explained the association between ethnicity and methylation in 51 additional loci. This left 263 loci associated with ethnicity after adjustment for ancestry where there was either no association between PC's 3–10 and methylation or the inclusion of these PC's did not affect the association between ethnicity and methylation. (*Supplementary file 1E*)

At these 314 loci, the median total variance accounted for by ethnicity, ancestry, and fine-scale substructure was 10.4% (IQR 6.6% to 16.1%), of which ethnicity explained a median of 1.7% (IQR 0.8% to 3.8%), ancestry explained a median of 2.9% (IQR 1.0% to 4.6%) and fine scale substructure explained a median of 3.4% (IQR 2.0% to 4.2%). Among the 263 CpG's whose association with

**Table 2.** Proportion of variance in methylation explained by ethnicity and ancestry. Numbers represent the median and interquartile range.

| Component | All CpG's associated with ethnicity (n = 916) | CpG's associated with ethnicity after adjusting for ancestry (n = 314) | CpG's whose association with ethnicity is explained by ancestry (n = 520) |
|---|---|---|---|
| Joint | 6.8% (4.5% to 10%) | 6.2% (4.4% to 8.8%) | 7.8% (5.3% to 11.1%) |
| Ethnicity | 1.7% (0.78% to 3.0%) | 3.5% (2.2% to 5.1%) | <1% |
| Ancestry | 4.2% (1.8% to 8.3%) | 1.8% (0.8% to 4.0%) | 6.6% (4.0% to 10.2%) |

ethnicity could not be explained by fine-scale substructure, ethnicity explained a median of 1.9% (IQR 1.0% to 4.0%; max 26.7%), ancestry explained 2.8% (IQR 1.0% to 6.2%), and fine scale ancestry explained 3.2% (IQR 1.9% to 4.7%).

As only 16 participants self-identified as 'Mixed Latino', we performed a sensitivity analysis to test the effect of excluding these participants from the analysis and only examining Puerto Ricans, Mexicans, and 'Other Latinos'. We found that excluding self-identified 'Mixed Latino' participants from the analysis did not significantly alter the results in most cases (*Supplementary file 1F*). Of the 916 CpG's associated with ethnicity at a genome-wide scale (p<1.6×10–7) in models including individuals self-identified as 'Mixed Ethnicity', 894 (97.5%) were still significant at a genome-wide scale when 'Mixed Latinos' were excluded. All but two of the CpG's that did not meet genome-wide significance were significant when correcting for 916 tests (p<5×10–5). In addition, an additional 290 CpG loci that did not meet genome-wide significance in the original analysis were significant at a genome-wide scale when self-identified 'Mixed Latinos' were excluded. While these loci did not meet genome-wide significance in the original analysis that included Mixed Latinos, they all had p-values lower than $2 \times 10^{-6}$. Thus we conclude that a sensitivity test excluding individuals of mixed Latino ethnicity did not significantly alter the conclusions.

We conclude that shared genetic ancestry explains much but not all of the association between ethnicity and methylation. Other, non-genetic factors associated with ethnicity likely explain the ethnicity-associated methylation changes that cannot be accounted for by genomic ancestry alone.

## Ethnic differences in environmentally-associated methylation sites

Methylation at CpG loci that had previously been reported to be associated with environmental exposures whose exposure prevalence differs between ethnic groups were tested for association with ethnicity in this study. A recent meta-analysis of maternal smoking during pregnancy, an exposure that varies significantly by ethnicity (*Oh et al., 2012*), identified associations with methylation at over 6000 CpG loci (*Joubert et al., 2016*). We found 1341 of 4404 that passed QC in our own study (30.4%) were nominally associated with ethnicity (p<0.05), which represented a highly significant (p<2×10$^{-16}$) enrichment. Using a Bonferroni correction for the 4404 loci tested, 126 maternal-smoking related loci were associated with ethnicity (p<1.1×10$^{-5}$), and 27 loci were among the 916 CpG's reported above as associated with ethnicity (*Supplementary file 1G*). Of these, 14 were among the 314 CpG's whose association with ethnicity could not be explained by ancestry and 12 were among the 263 CpG's whose association with ethnicity could not be explained by ancestry or fine-scale substructure. We also examined methylation loci from an earlier study of maternal smoking in Norwegian newborns (*Joubert et al., 2012*) as well as studies of diesel exhaust particles (*Jiang et al., 2014*) and exposure to violence (*Chen et al., 2013*). These results are supportive of our hypothesis that environmental exposures may be responsible for the observed differences in methylation between ethnic groups and are presented in *Supplementary file 1H*.

In an earlier study of maternal smoking in Norwegian newborns (*Joubert et al., 2012*) that identified 26 loci associated with maternal smoking during pregnancy, 19 passed quality control (QC) in our own analysis, and the association between methylation and ethnicity was found to be nominally significant (p<0.05)at 6 (31.6%) CpG loci. Adjusting for 19 tests (p<0.0026), cg23067299 in the aryl hydrocarbon receptor repressor (*AHRR*) gene on chromosome five remained statistically significant (*Supplementary file 1H*). These results suggest that ethnic differences in methylation at loci known to be responsive to tobacco smoke exposure *in utero* may be explained in part by ethnic-specific differences in the prevalence of maternal smoking during pregnancy.

We also found that CpG loci previously reported to be associated with diesel-exhaust particle (DEP) exposure (*Jiang et al., 2014*) were significantly enriched among the set of loci whose methylation levels varied between ethnic groups. Specifically, of the 101 CpG sites that were significantly associated with exposure to DEP and passed QC in our dataset, 31 were nominally associated with ethnicity (p<0.05), and five were associated with ethnicity after adjusting for 101 comparisons (p<0.005). Finally, we found that methylation levels at cg11218385 in the pituitary adenylate cyclase-activating polypeptide type I receptor gene (*ADCYAP1R1*), which had been associated with exposure to violence in Puerto Ricans (*Chen et al., 2013*) and with heavy trauma exposure in adults (*Ressler et al., 2011*), was significantly associated with ethnicity (p=0.02).

We also found 194 loci with a significant association between global genetic ancestry and methylation levels (after adjusting for ethnicity) at a Bonferroni corrected association p-value of less than

$1.6 \times 10^{-7}$ (*Figure 6* and *Supplementary file 1I*), including 48 that were associated with ethnicity in our earlier analysis. Of these significant associations, 55 were driven primarily by differences in African ancestry, 94 by differences in Native American ancestry, and 45 by differences in European ancestry. The most significant association between methylation and ancestry occurred at cg04922029 in the Duffy antigen receptor chemokine gene (DARC) on chromosome 1 (ANOVA p-value $3.1 \times 10^{-24}$) (*Figure 6B*). This finding was driven by a strong association between methylation level and global African ancestry; each 25 percentage point increase in African ancestry was associated with an increase in M-value of 0.98, which corresponds to an almost doubling in the ratio of methylated to unmethylated DNA at the site (95% CI 0.72 to 1.06 per 25% increase in African ancestry, $p=1.1 \times 10^{-21}$). There was no significant heterogeneity in the association between genetic ancestry and methylation between Puerto Ricans and Mexicans (p-het = 0.5). Mexicans have a mean unadjusted methylation M-value 0.48 units lower than Puerto Ricans (95% CI 0.35 to 0.62 units, $p=1.1 \times 10^{-11}$). However, adjusting for African ancestry accounts for the differences in methylation level between the two sub-groups (p-adjusted = 0.4), demonstrating that ethnic differences in methylation at this site are due to differences in African ancestry.

The distribution of methylation M-values at cg04922029 is tri-modal, raising the possibility that a SNP whose allele frequency differs between African and non-African populations may be driving the association. We therefore looked at the association between methylation at cg0422029 and ancestry at that locus. We found almost perfect correlation between methylation and African ancestry at the locus ($p=6 \times 10^{-162}$) (*Figure 7A*). Each African haplotype at the CpG site was associated with an increase in methylation M-value of 2.7, corresponding to a 6.5-fold increase in the ratio of methylated to unmethylated DNA per African haplotype at that locus. We then looked for SNPs within 10,000 base pairs of the CpG site that explained the admixture mapping association. We found that methylation at cg04922029 was significantly correlated with SNP rs2814778 (*Figure 7B*), the Duffy

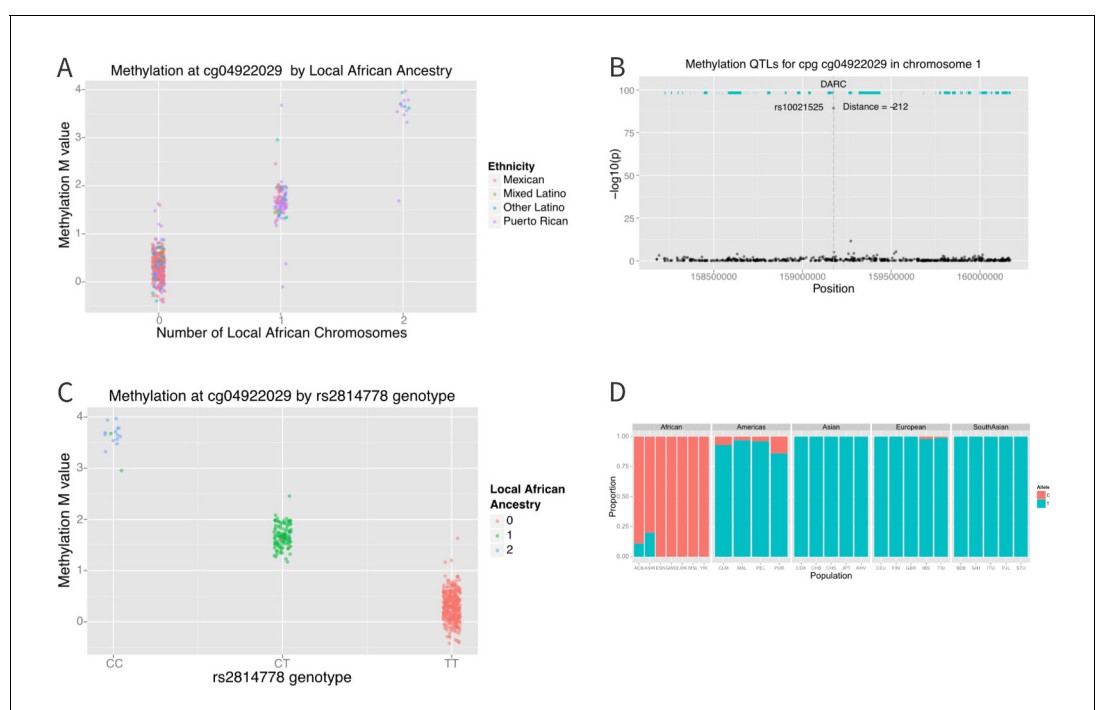

**Figure 7.** Association between local ancestry and methylation. (A) Association between cg04922029 on the DARC locus and African ancestry, color coded by ethnic group. There is near perfect correlation between the two. (B) Association between SNPs located within 1 Mb of cg04922029 and methylation levels at that CpG. (C) Association between rs2814778 (Duffy null) genotype and methylation at cg04922029, color coded by the number of African alleles present. There is near perfect correlation between genotype, ancestry and methylation at the locus. (D) Allele frequency of rs2814778 by 1000 Genomes population. The C allele is nearly ubiquitous in African populations and nearly absent outside of African populations and their descendants.

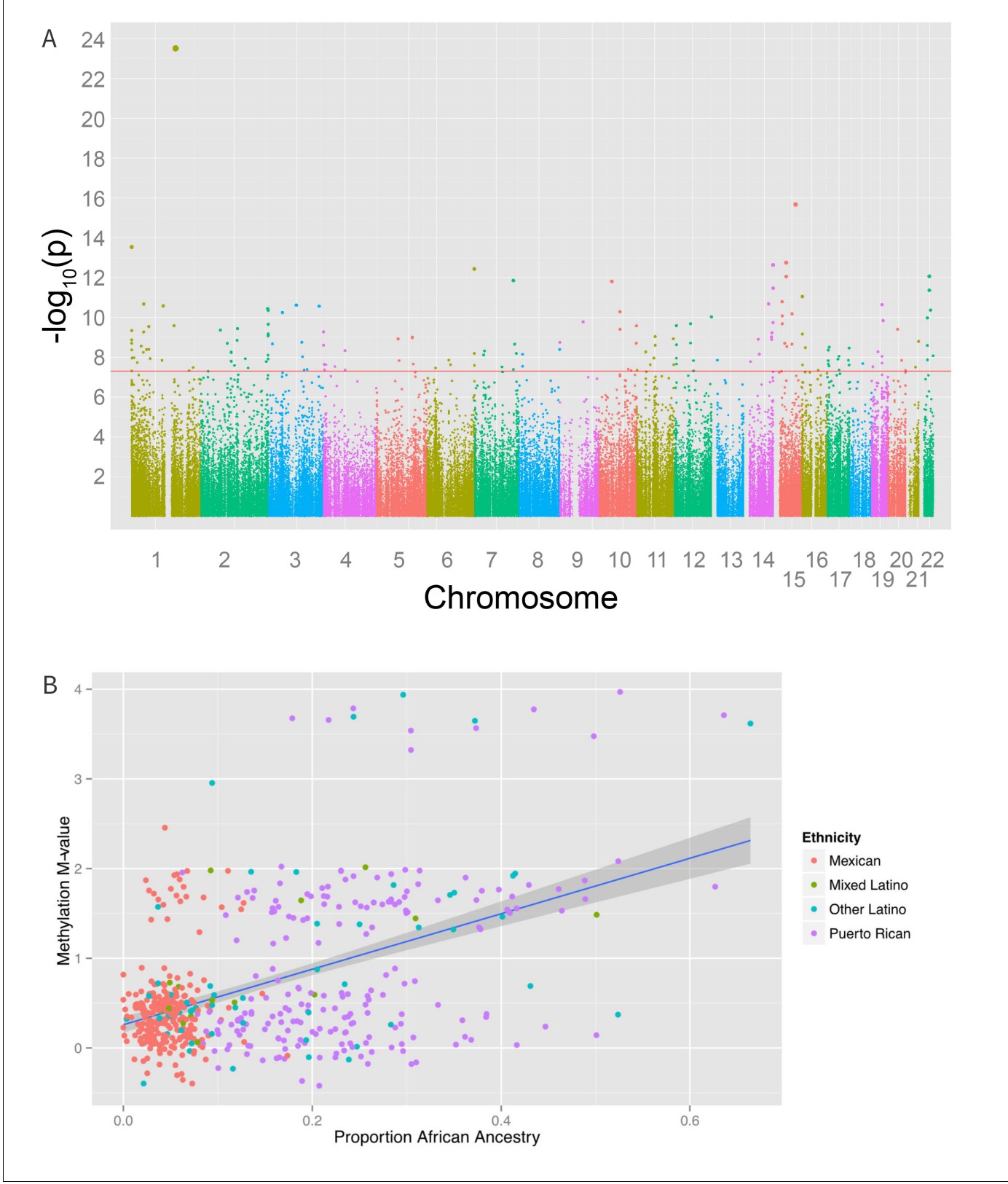

**Figure 6.** Associations between genomic ancestry and individual methylation loci. (A) Manhattan plot showing the associations between genomic ancestry and methylation at individual CpG loci. (B) Plot showing one such locus, cg04922029, and genomic African ancestry, showing a strong correlation between African ancestry and hypermethylation at that site.

null mutation, 212 base pairs away; each copy of the C allele was associated with an increase in M-value of 1.5, or a 2.9-fold increase in the ratio of methylated to unmethylated DNA (p=3.8×10$^{-90}$) (*Figure 7C*).

When we examined the effect of local ancestry at the other 194 CpG's we find that a substantial proportion of the effect of global ancestry on local methylation levels is due to local ancestry acting in –cis. Among the 194 CpG sites associated with global ancestry, local ancestry at the CpG site explained a median of 10.4% (IQR 3.0% to 19.4%) of the variance in methylation, accounting for a median of 52.8% (IQR 20.3% to 84.9%) of the total variance explained jointly by local and global ancestry (*Figure 8*).

## Discussion

In a diverse population of Latinos, we have shown that a substantial number of loci are differentially methylated between ethnic sub-groups. While genomic ancestry can explain the association between ethnicity and methylation at 66% of the 916 loci associated with ethnicity, factors other than shared ancestry that correlate with ethnicity, such as social, economic, cultural and environmental exposures account for the association between ethnicity and methylation at 34% (314/916) of loci.

We conclude that systematic environmental differences between ethnic subgroups likely play an important role in shaping the methylome for both individuals and populations. Loci previously associated with diverse environmental exposures such as *in utero* exposure to tobacco smoke (*Joubert et al., 2012*, *2016*), as well as diesel exhaust particles (*Jiang et al., 2014*) and psychosocial stress (*Chen et al., 2013*) were enriched in our set of loci where methylation was associated with ethnicity. Twenty-seven of the loci associated with maternal smoking during pregnancy in a large consortium meta-analysis (*Joubert et al., 2016*) were differentially methylated between Latino subgroups at a genome-wide significance threshold of $1.6 \times 10^{-7}$. Interestingly, this included both loci whose association persisted after adjustment for ancestry and fine-scale population substructure and are thus presumed to be due to environmental differences between ethnic groups and loci in which the association between ethnicity and methylation could be fully explained by genetically defined ancestry.

There are a number of plausible reasons for overlap between CpG's associated with ancestry and those associated with environmental exposure. It is possible that this represents a gene-environment interaction, and that individuals with certain genetic backgrounds are more susceptible to the effects

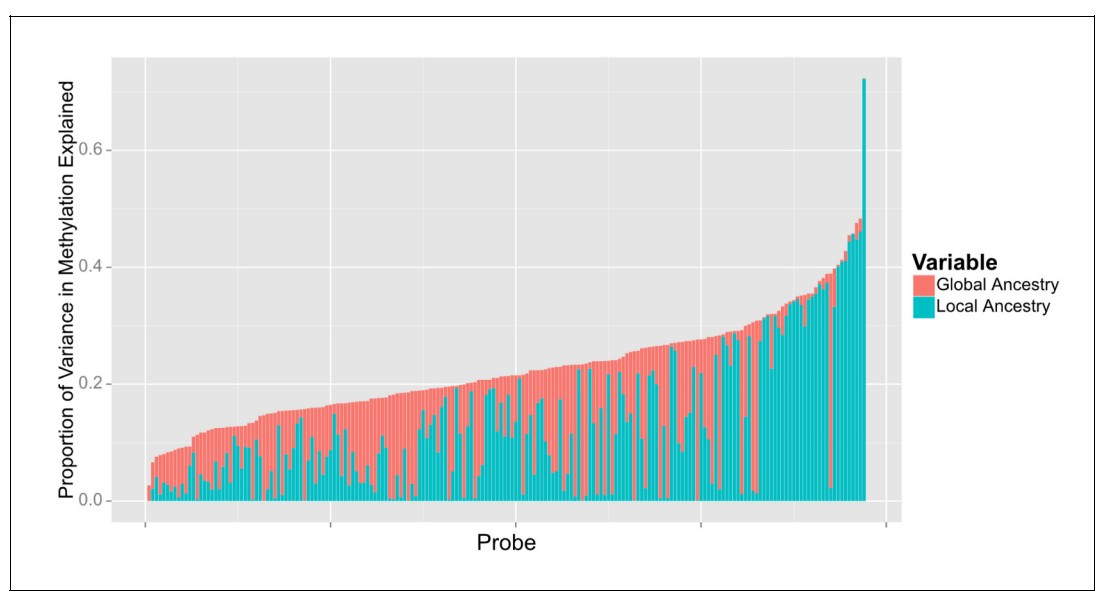

**Figure 8.** Relative proportion of variance in methylation explained by global and local ancestry across loci significantly associated with global ancestry.

of environmental exposures such as *in utero* tobacco smoke than those of other genetic backgrounds. It has been previously reported that Hispanic smokers with high Native American ancestry had reduced risk of methylation across 12 genes, suggesting an ancestry by smoking interaction (*Leng et al., 2013*). Because the majority of studies that comprised the consortium that identified differentially methylated regions enrolled participants of European descent, such interactions might not have been evident in their study.

It is also possible that environmental exposures correlate with ancestry and that participants with certain ancestral backgrounds may have been more exposed to *in utero* tobacco smoke than those of other backgrounds. Several studies have shown correlations between genetic ancestry and environmental exposures, including socioeconomic status (*Florez et al., 2011*), overweight and obesity (*Ziv et al., 2006*), and birth site and country of residence (*González Burchard et al., 2005*).

Though our analysis of global ancestry showed that a majority of the variance explained jointly by local and global ancestry can be traced to specific loci in the genome acting in –cis, a substantial proportion cannot. Some of the residual association between global ancestry and methylation may be due to genetic effects acting in –trans; however, the possibility that some of it may be due to environmental exposures correlating with global ancestry cannot be excluded. Thus, it is plausible that genomic ancestry is acting as a proxy for both genetic and environmental effects in our study. If this is the case, our study likely underestimates the degree to which environmental factors explain differential methylation between ethnic groups.

Finally, it is possible that our analysis identified DMRs that are independently modifiable by both genetic and environmental exposures. Thus, regions of the genome that are differentially methylated due to genetic polymorphisms may also be more susceptible to differential methylation due to environmental exposures.

Thus, inclusion of relevant social and environmental exposures in studies of methylation may help elucidate racial/ethnic disparities in disease prevalence, health outcomes and therapeutic response. However, in many cases, a detailed environmental exposure history is unknown, unmeasurable or poorly quantifiable, and race/ethnicity may be a useful, albeit imperfect proxy. However, if a comprehensive catalog of the effects of exposures can be compiled, it may be possible to use genome-wide methylation analysis as a biomarker of exposure long after the exposure has passed and can no longer be measured.

Our comprehensive analysis of high-density methyl- and genotyping from genomic DNA allowed us to investigate the genetic control of methylation in great detail and without the potential destabilizing effects of EBV transformation and culture in cell lines (*Grafodatskaya et al., 2010*). The strongest patterns of methylation are associated with cell composition in whole blood (*Lam et al., 2012*). However, the specific type of Latino ethnic-subgroups (Puerto Rican, Mexican, other, or mixed) is also associated with principal coordinates of genome-wide methylation.

Our approach has some potential limitations. It is possible that fine-scale population structure (sub-continental ancestry) within European, African, and Native American populations may contribute to ethnic differences in methylation, as we had previously reported in the case of lung function (*Moreno-Estrada et al., 2014*). However, despite the presence of additional substructure among the GALA II participants, PC's 3–10 explained the association between ethnicity and ancestry at only 51 loci. PCs from chip-based genotypes will not capture all forms of genetic variation. Clusters of ethnicity specific rare variants of large effect or strong ethnicity-specific selective sweeps in the last 8–12 generations (*Galanter et al., 2012*) could also give rise to methylation differences, but these are inconsistent with existing rare variant and selection analyses (*Hernandez et al., 2011*; *Tang et al., 2007*). Our models of genetic ancestry assumed a linear effect of ancestry on methylation, whereas a nonlinear association or other model misspecification could have led to incomplete adjustment for genetic ancestry, and thus, led to a residual association between ethnicity and methylation. However, when we added second and third order polynomials or cubic splines to our models, we found evidence for a nonlinear association between ancestry and methylation at only 25 and 26 loci, respectively, and it did not affect the association between ethnicity and methylation. Although it is impossible to account for all types of non-linearity and non-additivity (such as gene by gene or gene by environment interaction), our analysis suggests that non-linear effects are unlikely to be significant. Since our study was geographically diverse, recruiting participants at five recruitment sites in the United States and Puerto Rico, it is possible that systematic differences associated with site of recruitment might have influenced observed methylation differences between ethnic

groups. However, when we included recruitment site as a covariate, we found no significant effect on methylation independent of ethnicity.

The presence of a strong association between genetic ancestry and methylation raises the possibility that epigenetic studies can be confounded by population stratification, similar to genetic association studies, and that adjustment for either genetic ancestry or selected principal components is warranted. This possibility was first demonstrated in a previous analysis of the association between self-described race and methylation (*Barfield et al., 2014*). However, the study only evaluated two distinct racial groups (African Americans and Whites), while the present study demonstrates the possibility of population stratification in an admixed and heterogeneous population with participants from diverse Latino national origins. The tendency to consider Latinos as a homogenous or monolithic ethnic group makes any analysis of this population particularly challenging. Our finding of loci whose methylation patterns differed between Latino ethnic subgroups, even after adjusting for genetic ancestry, suggests that any analysis of these populations in disease-association studies without adjusting for ethnic heterogeneity is likely to result in spurious associations even after controlling for genomic ancestry. However, the methylation loci identified in this study, as well as studies of environmental exposures, could be particularly interesting loci for the study of biomedical outcomes, particularly those with disparate prevalence between racial/ethnic groups, such as asthma (*Barr et al., 2016*). If methylation loci associated with ethnicity or ancestry were shown to be associated with a biomedical outcome, it could help explain racial/ethnic disparities in disease.

In summary, this study provides a framework for understanding how genetic, social and environmental factors can contribute to systematic differences in methylation patterns between ethnic subgroups, even between presumably closely related populations such as Puerto Ricans and Mexicans. Methylation QTL's whose allele frequency varies by ancestry lead to an association between local ancestry and methylation level. This, in turn, leads to systematic variation in methylation patterns by ancestry, which then contributes to ethnic differences in genome-wide patterns of methylation. However, although genetic ancestry has been used to adjust for confounding in genetic studies, and can account for much of the ethnic differences in methylation in this study, ethnic identity is associated with methylation beyond the effects of shared genetic ancestry. This is likely due to social and environmental effects captured by ethnicity. Indeed, we find that CpG sites known to be influenced by social and environmental exposures are also differentially methylated between ethnic subgroups. These findings called attention to a more complete understanding of the effect of social and environmental variables on methylation in the context of race and ethnicity to fully understanding this complex process.

Our findings have important implications for the independent and joint effects of race, ethnicity, and genetic ancestry in biomedical research and clinical practice, especially in studies conducted in diverse or admixed populations. Our conclusions may be generalizable to any population that is racially mixed such as those from South Africa, India, and Brazil, though we would encourage further study in diverse populations, and likely has implications for all studies of diverse populations. As the National Institutes of Health (NIH) embarks on a precision medicine initiative, this research underscores the importance of including diverse populations and studying factors capturing the influence of social, cultural, and environmental factors, in addition to genetic ones, upon disparities in disease and drug response.

## Materials and methods

### Participant recruitment

All research on human subjects was approved by the Institutional Review Board at the University of California and each of the recruitment sites (Kaiser Permanente Northern California, Children's Hospital Oakland, Northwestern University, Children's Memorial Hospital Chicago, Baylor College of Medicine on behalf of the Texas Children's Hospital, VA Medical Center in Puerto Rico, the Albert Einstein College of Medicine on behalf of the Jacobi Medical Center in New York and the Western Review Board on behalf of the Centro de Neumologia Pediatrica), and all participants/parents provided age-appropriate written assent/consent. Latino children were enrolled as a part of the ongoing GALA II case-control study (*Oh et al., 2012*).

A total of 4702 children (2374 participants with asthma and 2328 healthy controls) were recruited from five centers (Chicago, Bronx, Houston, San Francisco Bay Area, and Puerto Rico) using a combination of community- and clinic-based recruitment. Participants were eligible if they were 8–21 years of age and self-identified as a specific Latino ethnicity and had four Latino grandparents. Asthma cases were defined as participants with a history of physician diagnosed asthma and the presence of two or more symptoms of coughing, wheezing, or shortness of breath in the two years preceding enrollment. Participants were excluded if they reported any of the following: (1) 10 or more pack-years of smoking; (2) any smoking within 1 year of recruitment date; (3) history of lung diseases other than asthma (cases) or chronic illness (cases and controls); or (4) pregnancy in the third trimester. Further details of recruitment are described elsewhere (*Oh et al., 2012*). Latino sub-ethnicity was determined by self-identification and the ethnicity of the their four grandparents. Due to small numbers, ethnicities other than Puerto Rican and Mexican were collapsed into a single category, 'other Latino'. Participants whose four grandparents were of discordant ethnicity were considered to be of 'mixed Latino' ethnicity.

Trained interviewers, proficient in both English and Spanish, administered questionnaires to gather baseline demographic data, as well as information on general health, asthma status, acculturation, social, and environmental exposures.

## Methylation

Genomic DNA (gDNA) was extracted from whole blood using Wizard Genomic DNA Purification Kits (Promega, Fitchburg, WI). A subset of 573 participants (311 cases with asthma and 262 healthy controls) was selected for methylation. Methylation was measured using the Infinium HumanMethylation450 BeadChip (Illumina, Inc., San Diego, CA) following the manufacturer's instructions.

1 µg of gDNA was bisulfite-converted using the Zymo EZ DNA Methylation Kit (Zymo research, Irvine, CA) according to the manufacturer's instructions. Bisulfite converted DNA was isothermally amplified overnight, enzymatically fragmented, precipitated, and re-suspended in hybridization buffer. The fragmented, re-suspended DNA samples were dispensed onto Infinitum HumanMethylation450 BeadChips and incubated overnight in an Illumina hybridization oven. Following hybridization, free DNA was washed away, and the BeadChips were extended through single nucleotide extensions with fluorescent labels. The BeadChips were imaged using an Illumina iScan system, and processed using the Illumina GenomeStudio Software.

Failed probes were identified using detection p-values using Illumina's recommendations. Probes on sex chromosomes and those known to contain genetic polymorphisms in the probe sequence were also excluded, leaving 321,503 probes for analysis. Raw data were normalized using Illumina's control probe scaling procedure. Beta values of methylation (ranging from 0 to 1) were converted to M-values via a logit transformation (*Du et al., 2010*).

## Genotyping

Details of genotyping and quality control procedures for single nucleotide polymorphisms (SNPs) and individuals have been described elsewhere (*Galanter et al., 2014*). Briefly, participants were genotyped at 818,154 SNPs on the Axiom Genome-Wide LAT 1, World Array 4 (Affymetrix, Santa Clara, CA) (*Hoffmann et al., 2011*). We removed SNPs with >5% missing data and failing platform-specific SNP quality criteria (n = 63,328), along with those out of Hardy-Weinberg equilibrium (n = 1845; $p<10-6$) within their respective populations (Puerto Rican, Mexican, and other Latino), as well as non-autosomal SNPs. Subjects were filtered based on 95% call rates and sex discrepancies, identity by descent and standard Affymetrix Axiom metrics. The total number of participants passing QC was 3804 (1902 asthmatic cases, 1902 healthy controls), and the total number of SNPs passing QC was 747,129. The number of participants with both methylation and genotyping data was 524.

## Ancestry and PCA analysis

GALA II participants were combined with ancestral data from 1000 Genomes European (CEU) and African (YRI) populations and 71 Native American (NAM) samples genotyped on the Axiom Genome-Wide LAT one array. A final sample of 568,037 autosomal SNPs with relevant ancestral data was used to estimate local and global ancestry. Global ancestry was estimated using the program ADMIXTURE (*Alexander et al., 2009*), with a three population model. Local ancestry at all

positions across the genome was estimated using the program LAMP-LD (*Baran et al., 2012*), assuming three ancestral populations.

Principal components for the genetic data were determined using the program EIGENSTRAT (*Patterson et al., 2006*).

## Statistical analysis

Using a variance in methylation m-value of 0.2 units, which corresponded to approximately the 90[th] percentile of the variance in m-value in our pilot data, we determined that in order to have an 80% power to detect a difference in mean methylation between the two major ethnic groups of 0.25 units, using a Bonferroni significance threshold of $1.6 \times 10^{-7}$ a sample, a sample size of 251 participants in each group was required. That total sample size of 502 participants gave us 80% power to detect correlations between ancestry and methylation of medium (Pearson r > 0.25) effect, meaning that we had 80% power to detect loci where ancestry accounted for at least 6.25% of the variance in methylation.

Unless otherwise noted, all regression models were adjusted for case status, age, sex, estimated cell counts, and plate and position. To account for possible heterogeneity in the cell type makeup of whole blood we inferred white cell counts using the method by Houseman et al (*Houseman et al., 2012*). Indicator variables were used to code categorical variables with more than two categories, such as ethnicity. In these cases, a nested analysis of variance (ANOVA) was used to compare models with and without the variables to obtain an omnibus p-value for the association between the categorical variable and the outcome. For analyses of dependent beta-distributed variables (such as African, European, and Native American ancestries), or cell proportion, k-1 variables were included in the analysis, and a nested analysis of variance (ANOVA) was used to compare models with and without the variables to obtain an k-1 degree of freedom omnibus p-value for the association between predictor (such as ancestry) and the outcome variable.

The Bonferroni method was used to adjust for multiple comparisons. For methylome-wide associations, the significance threshold was adjusted for 321,503 probes, resulting in a Bonferroni threshold of $1.6 \times 10^{-7}$. Analyses were performed using R version 3.2.1 (The R Foundation for Statistical Computing)(*R Core Team*) and the Bioconductor package version 2.13.

Multidimensional scaling of the logit transformed methylation data (M-values) was performed by first calculating the Euclidian distance matrix between each pair of individuals and then calculating the first 10 principal coordinates of the data (*Figure 2A*). We performed both a simple correlation analysis of these principal coordinates to demographic factors (age, sex, ethnicity), estimated cell counts and technical factors (batch, plate, and position) to identify factors that correlated with global methylation patterns [see *Figure 2B*]. In addition, we performed a multiple regression analysis of methylation principal coordinates by ethnicity and ancestry, adjusting for case status, age, sex, estimated cell counts, and plate and position (*Supplementary file 1A*).

We also sought to establish the extent to which global differences in methylation between Puerto Ricans and Mexicans could be explained by differences in ancestry between the two groups. We estimated the proportion of the ethnicity association that was mediated by genomic ancestry using the R package 'mediation' (*Tingley et al., 2014*) for methylation principal coordinates, which demonstrated a significant association with ethnicity.

We also sought to correlate ethnicity and methylation at a locus-specific level. We thus performed a linear regression between methylation at each CpG site and self-reported ethnicity (Mexican, Puerto Rican, Mixed Latino, and Other Latino), followed by a three degree of freedom analysis of variance to determine the overall effect of ethnicity on methylation We repeated the analysis excluding the 16 participants that were self-described as 'Mixed Latino', and tested for non-linearity in two ways: by adding second and third order polynomials to the model, and by adding a 3-degree of freedom cubic spline and comparing models with the non-linear terms to those without using a nested ANOVA. At loci where there was evidence for non-linearity, we tested whether ethnicity remained associated with methylation after adjusting for ancestry as well as the deviations from linearity. Finally, we tested for the presence of population sub-structure beyond that conveyed through ancestry by adding the genetic principal components 3–10 (PCs 1 and 2 were co-linear with ancestry with a coefficient of determination $R^2 > 0.998$) and comparing models with those PCs to those without. At loci where there was evidence for association between PC's 3–10 and methylation, we tested

whether ethnicity remained associated with methylation after adjusting for ancestry as well as the PC's 3–10.

We calculated the proportion of variance in methylation explained by ethnicity and genomic ancestry at each site where ethnicity was significantly associated with methylation. To do this, we fit a model that included both ethnicity and global ancestry as well as the confounders described above and calculated the proportion of variance explained by multiplying the ratio of the variance between predictors (ethnicity and genomic ancestry) and outcome (methylation) by the square of the effect magnitude (ß).

We also examined whether differences in methylation patterns by ethnicity could be associated with known loci that had previously been reported to vary based on common environmental exposures, including maternal smoking during pregnancy (*Joubert et al., 2012*), diesel exhaust particles (DEP) (*Jiang et al., 2014*), and exposure to violence (*Chen et al., 2013*). We have previously shown that exposure to these common environmental exposures or similar exposures varied by ethnicity within our own GALA II study populations (*Oh et al., 2012*; *Nishimura et al., 2013*; *Thakur et al., 2013*).

In addition, we examined the association between global ancestry and methylation across all CpG loci using a two-degree of freedom likelihood ratio test as well as by examining the association between individual ancestral components (African, European, and Native American) and methylation at each CpG site. At each site where methylation was significantly associated with genomic ancestry proportions, we determined the relative effect of global ancestry (θ, theta) and local ancestry (γ, gamma) in a joint model by calculating the proportion of variance explained as above.

To determine whether ancestry associations with methylation were due to variation in local ancestry, we correlated local ancestry at each CpG site with methylation at the site. Because ancestry LD is much stronger than genotypic LD, it is possible to accurately interpolate ancestry at each CpG site based on the ancestry estimated at the nearest SNPs (*Galanter et al., 2014*; *Rosenberg et al., 2010*). Measures of locus-specific ancestry were correlated with local methylation using linear regression. We performed a two-degree of freedom analysis of variance test evaluating the overall effect of all three ancestries as well as single-ancestry associations comparing methylation at a given locus with the number of African, European and Native American chromosomes at that CpG site.

## Acknowledgements

The authors acknowledge the families and patients for their participation and thank the numerous health care providers and community clinics for their support and participation in GALA II. In particular, the authors thank study coordinator Sandra Salazar; the recruiters who obtained the data: Duanny Alva, MD, Gaby Ayala-Rodriguez, Lisa Caine, Elizabeth Castellanos, Jaime Colon, Denise DeJesus, Blanca Lopez, Brenda Lopez, MD, Louis Martos, Vivian Medina, Juana Olivo, Mario Peralta, Esther Pomares, MD, Jihan Quraishi, Johanna Rodriguez, Shahdad Saeedi, Dean Soto, Ana Taveras. We also thank Sasha Gusev for helpful discussion. Computations in this manuscript were performed using the UCSF Biostatistics High Performance Computing System.

## Additional information

### Funding

| Funder | Grant reference number | Author |
| --- | --- | --- |
| National Institutes of Health | multiple | Joshua M Galanter<br>Christopher R Gignoux<br>Neeta Thakur<br>Harold J Farber<br>Rajesh Kumar<br>Max A Seibold<br>Esteban G Burchard<br>Noah Zaitlen |
| American Asthma Foundation | | Esteban G Burchard |
| Sandler Family Foundation | | Esteban G Burchard |

| | | |
|---|---|---|
| Tobacco-Related Disease Research Program | 24RT-0025 | Esteban G Burchard |
| Flight Attendant Medical Research Institute | | Esteban G Burchard |
| Hewett Fellowship | | Joshua M Galanter |
| Parker B. Francis Fellowship Program | | Neeta Thakur |
| American Thoracic Society | | Neeta Thakur |
| University of California, San Francisco | Chancellor's Research Fellowship | Christopher R Gignoux |
| University of California, San Francisco | Dissertation of the Year Fellowship | Christopher R Gignoux |
| Ernest S. Bazley Grant | | Pedro C Avila |

The funders had no role in study design, data collection and interpretation, or the decision to submit the work for publication.

## Author contributions
JMG, CRG, EGB, NZ, Conception and design, Analysis and interpretation of data, Drafting or revising the article; SSO, MP-Y, Analysis and interpretation of data, Drafting or revising the article; DT, Analysis and interpretation of data; NT, MAS, Drafting or revising the article; CE, Acquisition of data; DH, SH, Acquisition of data, Analysis and interpretation of data; HJF, RK, LNB, Recruited participants, Acquisition of data, Drafting or revising the article; PCA, EB-B, MAL, KM, DS, WR-C, JRR-S, Recruited participants, Acquisition of data

## Author ORCIDs
Joshua M Galanter, http://orcid.org/0000-0002-2561-6384
Christopher R Gignoux, http://orcid.org/0000-0001-9728-6567
Sam S Oh, http://orcid.org/0000-0002-2815-6037
Maria Pino-Yanes, http://orcid.org/0000-0003-0332-437X
Neeta Thakur, http://orcid.org/0000-0001-6126-6601
Esteban G Burchard, http://orcid.org/0000-0001-7475-2035

## Ethics
Human subjects: All research on human subjects was approved by the Institutional Review Board at the University of California and each of the recruitment sites (Kaiser Permanente Northern California, Children's Hospital Oakland, Northwestern University, Children's Memorial Hospital Chicago, Baylor College of Medicine on behalf of the Texas Children's Hospital, VA Medical Center in Puerto Rico, the Albert Einstein College of Medicine on behalf of the Jacobi Medical Center in New York and the Western Review Board on behalf of the Centro de Neumologia Pediatrica), and all participants/parents provided age-appropriate written assent/consent.

## Additional files

### Supplementary files
• Supplementary file 1. (A) Methylation principal components and ethnicity and ancestry. (B) Significant associations between ethnicity and methylation. (C) Effect of adding cubic spline ancestry terms to the association between ethnicity and methylation. (D) Effect of adding quadratic and cubic ancestry terms to the association between ethnicity and methylation. (E) Effect of adding genetic principal components 3–10 to the association between ethnicity and methylation. (F) Significant associations between ethnicity and methylation ($p<1.6\times10^{-7}$), and effect of adjustment for ancestry on the association of ethnicity and methylation, excluding participants of 'Mixed Latino' ethnicity. (G) Association of ethnicity and methylation in loci previously associated with maternal smoking during pregnancy. (H) Significant associations between ethnicity and methylation loci previously

associated with environmental exposures. (I) Significant associations between global ancestry and methylation.

## Major datasets

The following datasets were generated:

| Author(s) | Year | Dataset title | Dataset URL | Database, license, and accessibility information |
|---|---|---|---|---|
| Burchard EG | 2016 | Differential DNA methylation in Latino population | https://www.ncbi.nlm.nih.gov/geo/query/acc.cgi?acc=GSE77716 | Publicly available at the NCBI Gene Expression Omnibus (accession no. GSE77716) |
| Burchard EG | 2016 | Genes-Environment and Ancestry in Latino Americans | https://www.ncbi.nlm.nih.gov/gap/?term=phs001180 | Publicly available at NCBI dbGaP (accession no. phs001180.v1.p1) |

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
