## [Decision Letter]

Thank you for submitting your article "Differential methylation between ethnic sub-groups reflects the effect of genetic ancestry and environmental exposures" for consideration by *eLife*. Your article has been reviewed by Frank Johannes, Mark Shriver, and Marcus Nordborg (who is a member of our Board of Reviewing Editors) and the evaluation has been overseen by Mark McCarthy as the Senior Editor.

The reviewers have discussed the reviews with one another and the Reviewing Editor has drafted this decision to help you prepare a revised submission.

Summary:

A very clear paper that convincingly argues that self-described ethnicity explains methylation variation between individuals even after genetic ancestry has been taken into account, presumably because shared ethnicity implies shared environment. Although there is no demonstration that the findings have clinical importance, our consensus opinion is that the work is fundamentally interesting.

Essential revisions:

1) Principal component analysis of global patterns of methylation showed that when genetic ancestry was adjusted for, self-identified ethnicity remain significant for PC 6, suggesting that this measured contains other, non-genetic, factors. In an effort to identify specific DMRs that are associated with self-identified ethnicity you performed a EWA scan. This scan initially uncovered 916 genome-wide significant CpG sites. However, repeating this analysis after adjusting for genetic ancestry yielded only 314 significant associations. Hence, most of the initial 916 associations were due to genetic effects. Variance component of the remaining 314 sites showed that "genomic ancestry explained a median of 4.2% (IQR 1.8% to 8.3%) of the variance in methylation at these loci and accounts for a median of 75% (IQR 45.8% to 92%) of the variance in methylation explained jointly by ethnicity and ancestry". In the Abstract of the manuscript, you only report that "[…] shared genomic ancestry accounted for a median of 75.7% (IQR 45.8% to 92%) of the variance in methylation associated with ethnicity." Hence, we find the use of these results slightly misleading. Translating the 75% into effect sizes means that genetic ancestry and ethnicity jointly explain a median of 5.6% of the variance in DNA methylation at the 314 identified CpG sites, and that ethnicity explains merely 1.4% (median) of the variation. These effect sizes seem very modest, particularly the effects of ethnicity. It is also not clear to which extent these associations are independent. You should show the extent to which methylation scores at the 314 CpG sites are correlated.

2) You used Eigensoft to detect fine-scale sub-continental genetic ancestry. Eight of the Eigensoft PCs were used in a re-analysis of the 314 CpGs. Interestingly, an additional 51 loci seem to be explained by fine-scale genetic diversity; thus leaving 263 loci where ethnicity had significant marginal effects. You do not report the relative contribution of ethnicity from this re-analysis (i.e. are we still talking about 1.4% (median) variance explained, or is it now less?). This should probably be done.

3) Starting in subsection “Ethnic differences in environmentally-associated methylation sites”, you search for overlaps between the 916 DMRs from their initial EWA scan and CpGs sites previously shown to be associated with specific environmental variables. The motivation for this analysis seems to follow from the fact that (at least some of) these CpG sites are associated with ethnicity and are likely mediate uncharacterized environmental exposures. You do indeed find a significant enrichment with CpG sites shown to be correlated with maternal-smoking, diesel-exhaust particle exposure and heavy trauma exposure in adults. You conclude that "these results are supportive of our hypothesis that environmental exposures may be responsible for the observed differences in methylation between ethnic groups and are presented in Table S8". We don't understand why you go back to the 916 CpG sites for this analysis, considering that you already established that 916-263 = 653 of these are explained by genetic ancestry? Are you suggesting that there is genotype x environmental correlation? Furthermore, which of the 916 show enrichment with the environmental-associated CpG sites from previous studies? Are these mostly the 263 CpGs for which we have evidence that they are not fully explained by genetic ancestry? Or are we talking about many CpG sites that are strongly affected by genetic ancestry? The later would indeed imply that environmental associations of previous studies are mediated by genetic effects. If this is the case, the conclusions of this manuscript (and probably those of previous studies) would substantial change. This consideration should be thoroughly investigated and discussed.

4) You state in the Introduction that it is desirable to replace self-identified ethnicity (a social-construct) which biological constructs in biomedical research, because the latter are potentially better (and less biased) predictors of disease outcomes. Self-identified ethnicity subsumes genetic ancestry as well complex environmental variables such as social-economic status, diet, exposures to toxins, life-style choices, etc. Your earlier work already showed that genetic ancestry (inferred to genotype data) can be a better predictor of biomedical outcomes than self-identified ethnicity. The reason is that these self-reports poorly tag true genetic ancestry. Similarly, one can expect that these self-reports do not adequately tag specific environmental factors. If such environmental factors impact biomedical outcomes via their effects on DNA methylation it would be sensible to try to use DMRs directly as predictors of biomedical outcomes, in addition to genetic ancestry, even if these DMRs are not associated with self-identified ethnicity. Conversely, many of the 916 CpGs the authors identified in their EWAs analysis should be ideally be assessed for their impact on biomedical outcomes. It may well be that their effects are mostly neutral, or that their effect sizes are too small. This also applies to DMRs at CpGs that have been shown to be associated with maternal smoking. If this cannot be done, it should at least be discussed.

---

## [Author Response]

*Essential revisions:*

*1) Principal component analysis of global patterns of methylation showed that when genetic ancestry was adjusted for, self-identified ethnicity remain significant for PC 6, suggesting that this measured contains other, non-genetic, factors. In an effort to identify specific DMRs that are associated with self-identified ethnicity you performed a EWA scan. This scan initially uncovered 916 genome-wide significant CpG sites. However, repeating this analysis after adjusting for genetic ancestry yielded only 314 significant associations. Hence, most of the initial 916 associations were due to genetic effects. Variance component of the remaining 314 sites showed that "genomic ancestry explained a median of 4.2% (IQR 1.8% to 8.3%) of the variance in methylation at these loci and accounts for a median of 75% (IQR 45.8% to 92%) of the variance in methylation explained jointly by ethnicity and ancestry". In the Abstract of the manuscript, you only report that "[…] shared genomic ancestry accounted for a median of 75.7% (IQR 45.8% to 92%) of the variance in methylation associated with ethnicity." Hence, we find the use of these results slightly misleading. Translating the 75% into effect sizes means that genetic ancestry and ethnicity jointly explain a median of 5.6% of the variance in DNA methylation at the 314 identified CpG sites, and that ethnicity explains merely 1.4% (median) of the variation. These effect sizes seem very modest, particularly the effects of ethnicity. It is also not clear to which extent these associations are independent. You should show the extent to which methylation scores at the 314 CpG sites are correlated.*

Ancestry and ethnicity jointly explain a median of 6.8% of the variance in methylation (IQR 4.5% to 10.0%); the discrepancy from the reviewers’ calculation is due to the fact that the median of the sum is different than the sum of medians. The observed effect sizes appear modest in many cases because the reported numbers include all DMRs that were statistically associated with ethnicity, regardless of effect size. We note that in one CpG (cg09668627), ethnicity and ancestry jointly explained 38.5% of the variance in methylation and there were 17 CpG’s where ethnicity and ancestry jointly explain more 25% of the variance.

Moreover, because there is imperfect correlation between ethnicity and ancestry on the one hand, and the causative meQTL’s and presumed environmental factors on the other, the observed effects of ancestry and ethnicity on methylation are reduced. Thus, we find larger effect sizes in our analysis of the variance explained jointly by global and local ancestry. We find that among the 194 CpG sites that were associated with global genetic ancestry, local and global ancestry jointly explain a median of 21.4% of the variance (IQR 16.4% to 27.3%). Over 25% of the variance is explained in 65 of the 194 sites. The proportion of variance in methylation explained was as high as 72.2% at cg04922029 in the DARC gene (as one would expect, almost all of the variance is explained by local ancestry). However, when we examine the variance explained by ethnicity and global ancestry at cg04922029, we are able to explain a smaller proportion of the variance (30.7%, almost all of it captured by global ancestry). We hypothesize that a similar effect would be seen with environmental measures since ethnicity is an imperfect proxy for environmental exposure.

We would like to clarify that the variance component analysis was reported on all 916 CpG’s, not just the 314 that remained significant after adjusting for ancestry. We apologize for any confusion that arose from writing “at these loci”. At the 314 loci that remained associated with ethnicity, the median total variance explained jointly by ethnicity and ancestry was 6.2% (IQR 4.4% to 8.8%); ethnicity accounted for a median of 3.5% of the variance in methylation (IQR 2.2% to 5.1%) while ancestry accounted for a median of 1.8% (IQR 0.8% to 4.0%) and explained 32% (IQR 16.8% to 56.2%) of the total variance accounted for jointly by ethnicity and ancestry. Among the 520 CpG’s that were no longer associated with ethnicity when adjusted for ancestry, the median total variance explained jointly by ethnicity and ancestry was 7.8% (IQR 5.3% to 11.1%); ethnicity explains less than 1% of the variance in methylation, while ancestry explains 6.6% of the variance (IQR 4.0% to 10.2%), corresponding to a median of 88.0% of the variance jointly explained (IQR 75.6% to 95.4%). We added a table (Table 2) describing these findings and made reference to them in the Results section.

Generally, there was a moderate amount of correlation between the 314 methylation sites associated with ethnicity after correcting for ancestry. Among the 49,141 pairs of CpGs the median R^2^ was 0.044 (IQR 0.010 to 0.125). This is more correlation than was seen between 100 random methylation sites where median R^2^ was 0.012 (IQR 0.003 to 0.035), though we would expect greater correlation due to the common effect of ethnicity at the sites. We have made reference to this correlation in the Results section.

*2) You used Eigensoft to detect fine-scale sub-continental genetic ancestry. Eight of the Eigensoft PCs were used in a re-analysis of the 314 CpGs. Interestingly, an additional 51 loci seem to be explained by fine-scale genetic diversity; thus leaving 263 loci where ethnicity had significant marginal effects. You do not report the relative contribution of ethnicity from this re-analysis (i.e. are we still talking about 1.4% (median) variance explained, or is it now less?). This should probably be done.*

As suggested by the reviewers, we performed a similar analysis among the 314 CpG’s that we examined for higher order associations. Overall, among the 314 CpG’s, that remained associated with methylation after adjustment for ethnicity, the median total variance accounted for by ethnicity, ancestry, and fine-scale substructure was 10.4% (IQR 6.6% to 16.1%), of which ethnicity explained a median of 1.7% (IQR 0.8% to 3.8%), ancestry explained a median of 2.9% (IQR 1.0 to 4.6%) and fine scale substructure explained a median of 3.4% (IQR 2.0% to 4.2%). Among the CpG’s whose ethnicity association was not explained by fine-scale substructure, ethnicity explained a median of 1.9% (IQR 1.0% to 4.0%) and as high as 26.7%, ancestry explained 2.8% (IQR 1.0% to 6.2%), and fine scale ancestry explained 3.2% (IQR 1.9% to 4.7%). Note, however, that we would expect measures of the proportion of variance explained by fine scale population structure to be somewhat inflated due to the inclusion of 8 additional terms for each of the PC’s. This text was added to the Results section.

*3) Starting in subsection “Ethnic differences in environmentally-associated methylation sites”, you search for overlaps between the 916 DMRs from their initial EWA scan and CpGs sites previously shown to be associated with specific environmental variables. The motivation for this analysis seems to follow from the fact that (at least some of) these CpG sites are associated with ethnicity and are likely mediate uncharacterized environmental exposures. You do indeed find a significant enrichment with CpG sites shown to be correlated with maternal-smoking, diesel-exhaust particle exposure and heavy trauma exposure in adults. You conclude that "these results are supportive of our hypothesis that environmental exposures may be responsible for the observed differences in methylation between ethnic groups and are presented in Table S8". We don't understand why you go back to the 916 CpG sites for this analysis, considering that you already established that 916-263 = 653 of these are explained by genetic ancestry? Are you suggesting that there is genotype x environmental correlation? Furthermore, which of the 916 show enrichment with the environmental-associated CpG sites from previous studies? Are these mostly the 263 CpGs for which we have evidence that they are not fully explained by genetic ancestry? Or are we talking about many CpG sites that are strongly affected by genetic ancestry? The later would indeed imply that environmental associations of previous studies are mediated by genetic effects. If this is the case, the conclusions of this manuscript (and probably those of previous studies) would substantial change. This consideration should be thoroughly investigated and discussed.*

Our procedure for looking for enrichment among ethnicity associated loci identified in the Joubert paper is a bit different than what was understood by the reviewers. We first looked to see if there was an enrichment in nominal associations with ethnicity (p <.05) among all the CpG’s associated with in utero tobacco smoke that passed QC in our study (n = 4404). We found that 1341 (30.4% of them) were at least nominally associated with ethnicity, which was highly enriched (p = 2 x 10-16). We also found that 126 of those CpG’s were associated with ethnicity at a Bonferroni corrected p-value of 1.1x10-5 (corrected for 4404 comparisons). It was only then that we examined whether any were among the CpG’s associated with ethnicity at a genome-wide Bonferroni correction, and found that 27 of them were associated with ethnicity. Since we did not perform the ancestry adjustment of ethnicity genome-wide (we only performed it for the 916 loci that were significantly associated with ethnicity), we felt it would be most appropriate to look whether the in utero tobacco associated DMRs were within this larger group.

In light of the reviewers’ question, we examined the 314 DMRs that were associated with ethnicity even after adjusting for ancestry and found that 14 CpG’s were associated with in utero tobacco smoke in the Joubert et al. paper. We also examined the 263 CpG’s whose association with ethnicity could not be explained by ancestry or fine-scale population structure and found that 12 of them were associated with in utero tobacco smoke in the Joubert et al. paper. Neither of these results are significantly disproportionate to the proportion of CpG’s associated with (unadjusted) ethnicity.

There are a number of plausible reasons for overlap between CpG’s associated with ancestry and those associated with in utero tobacco smoke (or other environmental exposures). As the reviewers suggest, it is possible that this represents a gene-by-environment interaction, and that individuals with certain genetic backgrounds are more susceptible to the effects of in utero tobacco smoke than those of other genetic backgrounds. Leng et al. (AJRCCM, 2013) have showed that Hispanic smokers with high Native American ancestry had reduced risk of methylation across 12 genes, suggesting an ancestry by smoking interaction. Because the majority of studies in the consortium in the Joubert study enrolled participants of European descent, such interactions might not have been evident in their study.

It is also possible that environmental exposures correlate with ancestry and that participants with certain ancestral backgrounds may have been more exposed to in utero tobacco smoke than those of other backgrounds. Several studies have shown correlations between genetic ancestry and environmental exposures, including socioeconomic status (Florez et al., 2011), overweight and obesity (Ziv et al., 2006), and birth site and country of residence (Burchard et al., 2005). Though our analysis of global ancestry showed that a majority of the variance explained jointly by local and global ancestry can be traced to specific loci in the genome acting in -cis, a substantial proportion cannot. Although some of the residual association between global ancestry and methylation may be do to genetic effects acting in -trans, we cannot exclude the possibility that some of it may be due to environmental exposures correlating with global ancestry. Thus, it is plausible that genomic ancestry is acting as a proxy for both genetic and environmental effects in our study.

Finally, it is possible that our analysis identified DMRs that are independently modifiable by both genetic and environmental exposures. Thus, regions of the genome that are differentially methylated due to genetic polymorphisms may also be more susceptible to differential methylation due to environmental exposures.

At the reviewers’ suggestion, we reported the additional analysis in subsection “Ethnic differences in environmentally-associated methylation sites” and discussed the findings and their implications (Discussion section).

*4) You state in the Introduction that it is desirable to replace self-identified ethnicity (a social-construct) which biological constructs in biomedical research, because the latter are potentially better (and less biased) predictors of disease outcomes. Self-identified ethnicity subsumes genetic ancestry as well complex environmental variables such as social-economic status, diet, exposures to toxins, life-style choices, etc. Your earlier work already showed that genetic ancestry (inferred to genotype data) can be a better predictor of biomedical outcomes than self-identified ethnicity. The reason is that these self-reports poorly tag true genetic ancestry. Similarly, one can expect that these self-reports do not adequately tag specific environmental factors. If such environmental factors impact biomedical outcomes via their effects on DNA methylation it would be sensible to try to use DMRs directly as predictors of biomedical outcomes, in addition to genetic ancestry, even if these DMRs are not associated with self-identified ethnicity. Conversely, many of the 916 CpGs the authors identified in their EWAs analysis should be ideally be assessed for their impact on biomedical outcomes. It may well be that their effects are mostly neutral, or that their effect sizes are too small. This also applies to DMRs at CpGs that have been shown to be associated with maternal smoking. If this cannot be done, it should at least be discussed.*

We agree with the reviewers that one of the great promises for studies of methylation is their ability to tag environmental factors that may play a role in disease. Even in cases where the effect of an environmental exposure on disease is not directly mediated via methylation, the patterns of methylation that result from the exposure could be used as a lasting biomarker of exposure long after the exposure has passed. For example, in the study by Joubert et al. cited in our paper, measures of in utero exposure to tobacco persisted into childhood. Thus, as the reviewers point out, it would be sensible to perform an analysis examining associations between DMRs, particularly those known to be associated with ethnicity, ancestry and environmental exposures, and relevant biomedical outcomes (Discussion section).

Although it was beyond the scope of this paper to look at the association between DMRs and biomedical outcomes, we have performed a preliminary epigenome-wide analysis of asthma in this Latino population. Within that analysis, we find no evidence of enrichment in CpG’s associated with asthma among the 916 CpG’s associated with ethnicity in the current manuscript (p = 0.06 for enrichment, minimum p = 0.0009 for cg23702046, Bonferroni adjusted minimum p = 0.8). In the overall EWAS of asthma, we did find one CpG that was significantly associated with asthma, cg02458554 in chromosome 18 (p = 2.9 x 10^-7^). This CpG was not significantly associated with ethnicity (p = 0.4), but intriguingly, it was associated with genetic ancestry (p = 4.5 x 10^-4^). There did appear to be marginal enrichment in CpG’s associated with asthma among the results reported by Joubert et al. (p = 0.01), however, no individual CpG was associated with asthma when corrected for the number of comparisons (min p = 3.3 x 10^-5^, adjusted min-p = 0.1). These findings are awaiting replication and are outside the scope of this manuscript, but as recommended by the reviewers, we have expanded our Discussion along the lines noted.